# TST conversions and systemic interferon-gamma increase after methotrexate introduction in psoriasis patients

Vanessa Lucília Silveira de Medeiros[1,2¤a¤b]*, Fabiana Cristina Fulco Santos[3☯¤c], Lílian Maria Lapa Montenegro[3☯¤c], Maria da Conceição Silva[2¤b‡], Valdênia Maria Oliveira de Souza[2,4¤b¤d‡], Reginaldo Gonçalvez de Lima Neto[2¤b], Líbia Cristina Rocha Vilela Moura[2¤b], Vera Magalhães[2¤b]

**1** Dermatology Clinic, Clinics Hospital, Federal University of Pernambuco (UFPE), Recife, PE, Brazil, **2** Department of Tropical Medicine, Clinics Hospital, Federal University of Pernambuco (UFPE), Recife, PE, Brazil, **3** Department of Immunology, Laboratory of Immunoepidemiology, Aggeu Magalhães Research Center (CPqAM), Fiocruz, Federal University of Pernambuco (UFPE), Recife, PE, Brazil, **4** Laboratory of Immunopathology Keizo Asami-LIKA, Federal University of Pernambuco (UFPE), Recife, PE, Brazil

☯ These authors contributed equally to this work.
¤a Current address: Clínica de Dermatologia, Hospital das Clínicas, Cidade Universitária, Recife, PE, Brazil
¤b Current address: Aggeu Magalhães Research Center (CPqAM)–Fiocruz, Cidade Universitária, Recife, PE, Brazil
¤c Current address: Laboratory of Immunopathology Keizo Asami–LIKA, Cidade Universitária, Recife, PE, Brazil
¤d Current address: Centro de Ciências da Saúde, Departamento de Medicina Tropical, Hospital das clínicas, Bloco A—Térreo do HC/UFPE, Cidade Universitária, Recife, PE, Brazil
‡ MCS and VMOS also contributed equally to this work.
* vanessa.lucilia@ufpe.br

**Data Availability Statement:** All tables and results are available from https://sucupira.capes.gov.br/sucupira/public/consultas/coleta/

## Abstract

### Background

Tuberculosis screening in psoriasis patients is complex due to the immunological alterations associated with psoriasis, the presence of comorbidities, and the effect of immunosuppressive treatment. However, it is not established whether the results of screening tests are affected by these factors in psoriasis patients.

### Objectives

To determine whether there is a change in the results of the tuberculin skin test (TST) or the interferon-gamma release assay (IGRA) in psoriasis patients living in tuberculosis (TB)-endemic area after 12 weeks of methotrexate (MTX) treatment and to investigate the association of the test results with clinical and inflammatory markers.

### Methods

Forty-five patients were selected for a prospective single-arm self-controlled study and followed for at least 18 months. The TST, IGRA, Psoriasis Area and Severity Index (PASI), and inflammatory factors (erythrocyte sedimentation rate (ESR), C-reactive protein, interferon-gamma (IFN-γ), and tumor necrosis factor-alpha levels), were determined before and

trabalhoConclusao/viewTrabalhoConclusao.jsf?
popup=true&id_trabalho=6291465.

**Funding:** The author(s) received no specific
funding for this work.

**Competing interests:** The authors have declared
that no competing interests exist.

after 12 weeks of oral 15 mg per week MTX administration and compared. The associations
between the IGRA and TST results were verified before and after treatment according to
inflammatory factors and clinical characteristics (age, blood glucose, weight, body mass
index, disease duration, and PASI).

## Results

We collected data on 25 patients who completed the full course of therapy and the follow-
up. None of the patients developed TB. TST positivity was significantly elevated at week 12
(25% baseline vs 44% at week 12, P < 0.037). Three IGRAs followed the TST conversions.
There was no difference between TST and IGRA pre- or posttreatment. Serum IFN-γ
increased significantly in week 12 (15.95 pg/ml baseline vs 18.82 pg/ml at week 12, P <
0.005) and tended to be higher among TST-positive patients (P = 0.072). The baseline
IGRA was associated with a higher ESR (P = 0.038). None of the test results were associ-
ated with clinical characteristics.

## Conclusions

In addition to the classic booster effect, TST conversions in patients using MTX can occur
due to an increase in IFN-γ. However, it is not possible to exclude true TST conversions.
Therefore, other diagnostic methods, like IGRA or chest tomography, should be used when
the TST has intermediate results.

## Introduction

Nearly 3% of the total population may exhibit psoriasis during their lives [1]. This is an
immune-mediated T helper (Th) 17, Th22, and tumor necrosis factor-alpha (TNF-α) disease
with systemic and cutaneous inflammatory repercussions [2]. There is an extreme spectrum of
severity, with the more severe cases requiring systemic immunosuppressive therapy [3]. The
commonly used treatment options are methotrexate (MTX), ciclosporin, and, more recently,
TNF-α inhibitors and interleukin (IL) inhibitors [3].

MTX is a dihydrofolate reductase inhibitor that prevents folic acid activation, limits DNA
formation and cellular division, and has cytotoxic effects, leading to cell death by apoptosis
[4, 5]. MTX is a consolidated and highly available option for treating psoriasis, reducing the
number of keratinocytes in plaques of psoriasis, and decreasing inflammation [6]. However,
the precise underlying mechanism of action of MTX in psoriasis is not clearly understood [4,
5].

Treatment with biological drugs, especially TNF-α inhibitors, increases the risk of reactivat-
ing latent tuberculosis infection (LTBI) in psoriasis patients [7]. Psoriasis patients often have
several comorbidities, such as diabetes and obesity, which could increase the risk of LTBI [8].
Therefore, they must be screened for LTBI before the use of immunobiological medications [7,
9]. However, this task proves to be extremely challenging because there is no gold standard test
for LTBI. Many factors, such as vaccination, age, and migratory status, make obtaining the cor-
rect diagnosis even more difficult, especially in endemic TB areas [10–14].

Another barrier to LTBI diagnosis is determining the correct timing of testing [9].
Increased cytokine expression in psoriasis may cause incorrect screening test results if the test
is performed before immunosuppressive therapy [10]. However, numerous psoriasis patients

have already started treatment with an immunosuppressor such as MTX when they are referred for LTBI testing, which can alter the test results [9, 10].

To date, no prospective study has addressed the role of MTX therapy on tuberculin skin test (TST) or interferon-gamma release assay (IGRA) results among psoriasis patients living in TB-endemic areas [10–15]. Data on the relationship between cytokines of the Th1 pathway, the inflammatory rates, the clinical characteristics, and the results of the screening tests are also missing [10–15].

Therefore, this study aimed to investigate whether 12-week MTX use can modify the results and the agreement of the TST test and IGRA assays (QuantiFERON-TB Gold) in systemic treatment-naive psoriasis patients vaccinated with the bacillus Calmette-Guérin (BCG) vaccine who were born and lived in a TB-endemic region. Additionally, the study also explored associations of the TB screening tests with clinical and inflammatory factors.

## Materials and methods

### Study design

This study was a prospective, quasi-experimental, single-arm, self-controlled study associated with a cohort. The study period was between April 2015 and February 2018. The inclusion period was between April 2015 and August 2016. All patients were followed for a minimum period of 18 months after inclusion. After the end of the study, the patients continued the follow-up in the outpatient dermatology clinics.

### Ethics statement

The study was approved by the ethics committee of the Federal University of Pernambuco with trial number 42859015.0.0000.5208-CAAE on 01/04/2015. The trial was registered at http://www.ensaiosclinicos.gov.br/rg/RBR-2mt78b/. The study was registered after the enrolment of the patients because the site allows registration before or after the inclusion of participants according to the protocol approved by the ethics committee. The study followed the principles of the Declaration of Helsinki. The patients were included after signing the consent form. The authors confirm that all ongoing and related trials for this drug/intervention are registered.

### Sample and eligibility criteria

The inclusion criteria called for patients over 18 years of age, with moderate to severe plaque psoriasis, living in the northeast region of Brazil, referred by dermatologists to initiate the follow-up in the Dermatology Clinic of the Federal University of Pernambuco.

The exclusion criteria were the absence of a BCG vaccination scar, pregnant women, breastfeeding women, previous use of systemic treatment for psoriasis, previous use of immunosuppressants, use of topical medications for psoriasis in the last 6 months, use of drugs aggravating psoriasis, personal history or signs or symptoms of TB, chest X-ray compatible with lung disease other than LTBI, previous malignancy, positive serology for HIV, hepatitis B or C, liver enzyme elevation, renal function impairment, anemia, lymphopenia ($< 1500$ lymphocytes), platelet count $< 100,000$, or leukopenia $< 3500$ leukocytes.

The sample size was calculated based on the reduction of the Psoriasis Area and Severity Index (PASI) by MTX treatment for comparing paired differences. To detect an effect size of 0.65 between paired data with a power of 80% and a two-sided significance level of 5%, the necessary sample size for the paired sample T-test was 22 pairs.

Forty-five patients were referred to initiate follow-up during the study period. The patients underwent general laboratory tests in the hospital laboratory before treatment had been initiated, including complete blood count, renal function, liver enzymes, erythrocyte sedimentation rate (ESR), C-reactive protein (CRP), a pregnancy test for women, and serology for HIV and hepatitis B and C. Chest radiography was performed and read in the radiology department of the hospital.

Twelve patients were excluded due to laboratory alteration (ten for liver enzyme elevation, two for renal impairment, one for lymphopenia, and one for anemia); one was excluded due to pregnancy, one was excluded due to HIV, and two were excluded due to failure to return the exams. Three stopped taking the medication during the follow-up, and one was excluded due to coagulation of the blood sample. There were no exclusions due to adverse effects. Twenty-five patients completed the research protocol and the 18-month follow-up.

## Data collection

The demographic data collected included sex, age in years, disease duration in years, diabetes (use of medication for type 2 diabetes or glycemia > 126 mg/dl), place of residence (urban or rural), and contact with cattle.

All patients were born and lived in a high-risk area for TB. A standardized risk questionnaire was used to evaluate the presence of active TB and other risk factors for LTBI (cough, asthenia, fever, night sweats, anorexia, weight loss, history of drug abuse, previous contact with a carrier of tuberculosis). The patients were examined, and their weight, height, BMI, and PASI were determined.

## Laboratory tests

Blood samples were obtained by direct venipuncture by trained staff. For the QuantiFERON-TB Gold test, one milliliter of blood was collected in each of three tubes separately according to the sequence in the manufacturer's instructions (Cellectis, Melbourne, Australia). For the interferon-gamma (IFN-γ) and TNF-α tests, 3 ml of blood was collected in a tube with citrate. Samples were immediately sent to the laboratory, incubated within one hour after collection, and frozen.

After the collection of blood, TST was performed by trained staff using the Mantoux technique. Two units of purified protein derivative (PPD, RT23, Statens Serum Institut, Copenhagen, Denmark) were injected into the anterior surface of the left forearm. The result of TST was recorded in millimeters, and photographs were obtained at 72 hours. On the day of the TST read, the patients were instructed to intake 7.5 mg of MTX twice in a single day for a total of 15 mg per week without dose progression and 5 mg of folic acid per week. MTX treatment was started only after obtaining the result of the first TST.

Patients were advised about the most frequent adverse effects of MTX and the importance of continuing their medication. They were reevaluated for adverse effects 30 days after starting their medication. If adverse effects were noticed earlier, patients were to attend a consultation before the 30-day reevaluation. Patients who discontinued the medication or skipped doses for any reason were excluded.

The same TST protocol was repeated after 12 weeks of MTX use. At the second visit, TST was applied in the right forearm. Afterward, patients were reassessed every 3 months for clinical and laboratory evaluation. They continued to use MTX 15 mg per week until the treatment ceased to elicit a response, the patients presented significant gastrointestinal intolerance, laboratory changes requiring adjustment of the dosage, the route of administration was changed, or the medication was replaced with another.

After the end of the inclusion period, the QuantiFERON-TB Gold test and the cytokine (INVITROGEN, CA, EUA) measurements were analyzed using the Enzyme-Linked Immuno-sorbent Assay (ELISA) technique with standardized kits according to the manufacturer's instructions. All tests were performed together.

## Interpretation of results

The TST results were analyzed in two ways. First, the TST result was determined to be negative (−) when the size was < 5 mm, indeterminate (I) at ≥ 5 to < 10 mm, and positive (+) at ≥ 10 mm. Second, the TST result was determined to be negative when the induration was < 5 mm and positive when it was ≥ 5 mm.

The IGRA results were also analyzed in two ways. The first analysis was conducted according to the manufacturer's instructions for positive (+), negative (−), and indeterminate (I) results. As with the TST, which can have its cutting pointer rated at 5mm, we carried out a second IGRA analysis in which the indeterminate results were considered positive [16, 17].

The limits of detection for cytokines were defined according to the manufacturer's instructions (INVITROGEN, CA, EUA). The lower limit was 4 pg/ml for IFN-γ and 3.0 pg/ml for TNF-α. Results below the cutoff points were considered equal to zero. Missing data for any variable were set to zero.

## Statistical analyses

The statistical analysis was performed using the IBM Statistical Package version 23.0 (IBM Corp., Armonk, NY). The sample demographic and clinic characteristics are presented descriptively through absolute and percentage frequencies. Cohen's kappa test was used to assess the agreement between each test and between IGRA and TST results before and after treatment. McNemar's test was used to verify the change in the test results classified into two categories (positive/negative) and in comparisons of tests. The McNemar-Bowker test was used when the results were classified into three categories (positive/negative/indeterminate). For numeric continuous variables, the normality was tested using the Shapiro-Wilk test. To compare the mean of paired continuous variables (PASI, ESR, CRP, TNF-α, and IFN-γ), Student's t-test for paired variables was used for variables with a normal distribution, and the Wilcoxon test for paired data was used for variables with a non-normal distribution. To compare the TST and IGRA results with the means of numerical continuous clinical (age, disease duration, weight, blood glucose, BMI) and inflammatory variables (PASI, ESR, CRP, TNF-α, and IFN-γ), Student's t-test was performed for variables with a normal distribution, and the Mann-Whitney test was used for variables with a non-normal distribution. Statistical significance was defined as a P-value < 0.05. Missing data were excluded from the analysis.

## Results

The sample consisted of 14 women and 11 men. The sample baseline characteristics are described in Table 1. No patients developed tuberculosis during the study period. Any adverse effect was observed during the study period.

At baseline, five TST results were > 5 mm, and four were > 10 mm. Two IGRA results were positive, and three were indeterminate (Table 2). Five negative TST results converted to positive, and three baseline positive test results increased in size, resulting in 11 TST results > 5 mm and six results > 10 mm at 12 weeks. Three patients (4, 17, and 22) exhibited conversion of negative IGRAs at the same time as TSTs. Three indeterminate IGRA results reverted to negative in the second test, and one was converted to positive. Only four patients had contact with cattle; two were IGRA−/TST− both times (patients 6,14), one patient was

**Table 1. Baseline demographic and clinical characteristics of the patients enrolled in the study.**

| Characteristic | n | % |
|---|---|---|
| | 25 | 100.00 |
| Sex: | | |
| Female | 14 | 56.0 |
| Male | 11 | 44.0 |
| Age group (years): | | |
| 20 to 29 | 6 | 24.0 |
| 30 to 59 | 11 | 44.0 |
| 60 to 75 | 8 | 32.0 |
| Duration of psoriasis (years): | | |
| 1 to 2 | 6 | 24.0 |
| 3 to 9 | 8 | 32.0 |
| ≥ 10 | 11 | 44.0 |
| Diabetes: | | |
| Yes | 6 | 24.0 |
| No | 19 | 76.0 |
| Place of residence: | | |
| Urban region | 23 | 92.0 |
| Rural region | 2 | 8.0 |
| Contact with cattle: | | |
| Yes | 4 | 16.0 |
| No | 21 | 84.0 |
| Previous contact with a carrier of tuberculosis: | | |
| Yes | 9 | 36.0 |
| No | 16 | 64.0 |
| Chest X-ray suggestive of LTBI: | | |
| Yes | 3 | 12.0 |
| No | 22 | 88.0 |
| Percentage of PASI reduction: | | |
| < 50 | 3 | 12.0 |
| 50 to 75 | 5 | 20.0 |
| > 75 | 17 | 68.0 |

PASI, psoriasis area and severity index.

TST−/IGRA Indeterminate both times (25), and one patient (17) was negative on both tests the first time but converted to positive on both tests (Table 2).

The TST results became significantly positive on the second test in both analyses (P = 0.037/P = 0.032). IGRA results did not show a pattern of change (P = 0.135/1.0). The TST had a greater agreement (Moderate) before and after treatment than the IGRA (Fair), regardless of the number of categories. The kappa value remained in the same range for the IGRA (κ = 0.27/0.34) and TST (κ = 0.57/0.46) whether they were analyzed with 3 or 2 categories. The agreement between the IGRA and TST was the same before and after treatment (κ = 0.34/0.32). There was no difference between the tests before (P = 1.00) or after (0.289) treatment (Table 3).

Comparison of the means and medians before and after treatment revealed that CRP levels (P = 0.022) and PASI scores (P < 0.001) decreased significantly after treatment, while the IFN-γ level (P = 0.005) increased. There were missing data for TNF-α and ESR. A decrease in the

**Table 2. TST and IGRA results before and after 12 weeks of treatment with methotrexate, excluding 12 patients who tested negative both times.**

| Patient | TST 1 | TST 2 | IGRA 1 | IGRA 2 |
|---|---|---|---|---|
| 3 | 22 | 22 | POSITIVE | POSITIVE |
| 4 | 0 | 6 | NEGATIVE | INDETERMINATE |
| 5 | 12 | 22 | NEGATIVE | NEGATIVE |
| 12 | 5.5 | 9 | NEGATIVE | NEGATIVE |
| 13 | 11 | 11 | INDETERMINATE | NEGATIVE |
| 16 | 0 | 7 | NEGATIVE | NEGATIVE |
| 17 | 0 | 8 | NEGATIVE | POSITIVE |
| 18 | 22 | 22 | POSITIVE | POSITIVE |
| 22 | 5 | 25 | NEGATIVE | POSITIVE |
| 25 | 0 | 0 | INDETERMINATE | INDETERMINATE |
| 26 | 0 | 7 | NEGATIVE | NEGATIVE |
| 28 | 0 | 0 | INDETERMINATE | NEGATIVE |
| 29 | 0 | 20 | INDETERMINATE | NEGATIVE |

TST, tuberculin skin test; IGRA, interferon-gamma release assay.

ESR and an increase in TNF-α were observed after treatment, but these changes were not significant (Table 4).

There was no association between the TST or IGRA results before or after treatment and age, disease duration, weight, BMI, blood glucose, CRP, or absolute PASI (absolute and percentage) before or after treatment (S1–S4 Tables). Before treatment, there was an association between positive IGRA and higher levels of ESR (P = 0.038) (S1 Table). After treatment, there was no longer an association with ESR (P = 0.443) (S3 Table). After treatment, there was a trend toward an IFN-γ increase (P = 0.072) in the TST-positive group (S4 Table).

## Discussion

Many studies report a discordance between LTBI screening tests in patients with psoriasis [10–14]. The TST is usually considered to be responsible for this because it would be easily affected by clinical and epidemiological factors than the IGRA, leading to false-positive results. The exception is immunosuppressed and elderly individuals, who can have false-negative tests

**Table 3. Concordance between tuberculosis screening tests before and after 12 weeks of treatment with methotrexate.**

| Variable | Observed | | Agreement | | McNemar test |
|---|---|---|---|---|---|
| | n | % | k value | 95% CI | |
| Total | 25 | 100 | | | |
| TST/TST 3 categories | 20 | 80.0 | 0.57 | 0.27 to 0.87 | P = 0.037* |
| TST/TST 2 categories | 18 | 72 | 0.46 | 0.19 to 0.73 | P = 0.032* |
| IGRA/IGRA 3 categories | 18 | 72.0 | 0.27 | 0.14 to 0.69 | P = 0.135 |
| IGRA/IGRA 2 categories | 18 | 72.0 | 0.34 | -0.03 to 0.71 | P = 1.000 |
| TST/IGRA before | 19 | 76.0 | 0.34 | -0.08 to 0.76 | P = 1.000 |
| TST/IGRA after | 17 | 68.0 | 0.32 | -0.03 to 0.68 | P = 0.289 |

TST, tuberculin skin test; IGRA, interferon-gamma release assay. TST 3 categories; negative when t < 5 mm, indeterminate at ≥ 5 to < 10 mm, and positive at ≥ 10 mm. TST 2 categories: negative < 5 mm and positive when ≥ 5 mm. IGRA 3 categories: positive, negative, and indeterminate results. IGRA 2 categories: positive (positive and indeterminate results) and negative (negative results).
(*) P < 0.05.

**Table 4. Differences between CRP, ESR, IFN-γ, TNF-α, and PASI before and after 12 weeks of MTX treatment.**

| Variable | Before | After | P-value |
|---|---|---|---|
| CRP (mg/dl): Median (P25; P75) | 0.54 (0.12; 3.00) | 0.40 (0.14; 0.90) | P [(1)] = 0.022 * |
| ESR (mm): Median (P25; P75) | 13.00 (6.25; 22.25) | 7.50 (4.00; 12.75) | P [(1)] = 0.086 |
| IFN-γ (pg/ml): Mean ± SD | 15.95 ± 7.89 (12.68) | 18.82 ± 9.17 (16.44) | P [(2)] = 0.005 * |
| TNF-α (pg/ml): Median (P25; P75) | 0.00 (0.00; 1.97) | 3.68 (0.16; 6.61) | P [(1)] = 0.224 |
| PASI: Median (P25; P75) | 13.20 (10.30; 18.90) | 2.00 (0.95; 4.40) | P [(1)] < 0.001 * |

ESR, erythrocyte sedimentation rate; CRP, C-reactive protein; IFN-γ, interferon-gamma; TNF-α, tumor necrosis factor-alpha; PASI, psoriasis area, and severity index.

(*) $P < 0.05$ (1) Wilcoxon test for paired data. (2) Student's t-test for paired data.

[10–12]. In this study, a significant number of TSTs showed positive conversions despite the use of MTX. However, the IGRA presented changes that are not significant but should not be ignored. Studies with serial IGRAs showed changes in the test results in many patient profiles, including psoriasis, with the cause not being completely understood, but may be related to a low release of IFN-γ [18–21].

The known factors responsible for the discordant positive TST results are the following: having lived in an endemic area for TB, age, possible cross-reaction with *Mycobacterium bovis* (*M. bovis*) [11–14], and BCG vaccination [22–24]. In endemic areas, BCG is administered at birth, and it is not possible to exclude the environmental presence of *Mycobacterium tuberculosis* (*Mtb*) [25, 26]. Therefore, it is necessary to study the test results separately in these populations.

A possible relationship between test results and age was not found. BCG vaccine is also the reason why the TST has a higher positivity in patients under 15 years old [22, 23], while there is a possibility of the IGRA exhibiting a decrease in sensitivity in children aged under 2 years [24]. Therefore, only patients older than 18 years were included. The TSTs-negative tests could be more frequent in the elderly due to a loss of this immune response [26]. However, in comparative studies, the results of the TST and IGRA have been similar in the elderly [27, 28], consistent with our results. The number of participants who had previous contact with cattle was low, and the LTBI screening test results were mostly negative, reducing the possibility of a cross-reaction *M. bovis* with the TST [12, 13].

When it is not possible to identify epidemiological factors to justify the disagreements between the screening tests it is considered the possibility of interference by the inflammatory process of psoriasis [10–15, 29–31]. Psoriasis is a unique disease with cutaneous and systemic immunologic alterations [1, 15], and its relationship to the screening tests can be assessed in many ways, such as by the PASI at the time of the test, disease duration, comorbidities, inflammatory markers such as ESR, CRP, and cytokines of the Th17 and Th1 pathways. The cytokine levels of the Th17 pathway have not yet been evaluated. To date, no association has been found between IGRA or TST outcomes and the PASI [10, 29]. Studies in other populations did not find an association of the tests with diabetes and obesity [8]; neither did this study. The association between the IGRA and ESR before treatment indicates the possibility of interference of some inflammatory or infectious process in this test.

Another possible factor related to the disagreement in test results is immunosuppression [10, 14]. Each drug has a peculiar mechanism of action in modulating the Th1 and Th17 pathways and should be evaluated individually [15]. The most obvious reason is that the immunosuppressive effect of MTX on lymphocytes would increase the number of negative screening tests, as in other conditions such as HIV [13, 14, 32]. Lymphopenia is associated with negative IGRAs and could be induced by MTX [33]. Patients did not have lymphopenia at the time of testing, and there was no increase in negative IGRAs [11, 16, 34].

A significant increase in positive TST results and IFN-γ was observed after MTX treatment. The missing data for TNF-α were probably responsible for the lack of a significant increase in their results. Tavast et al. [11] found an association between immunosuppression and positive TST results, and one multicentric study found high rates of TST positivity during MTX therapy [35]. Experimental studies observed that mononuclear cell death increases IFN-γ and TNF-α due to the cytotoxic effect of MTX [36–38]. These cytokines are responsible for macrophage activation and the containment of *Mtb* [29]. Following the cutaneous cell death in psoriasis plaques induced by MTX, patients were again exposed to tuberculin and presented a greater capacity to contain the protein, which increased TST. The tendency toward higher levels of IFN-γ in positive TST patients may be related to this fact.

However, not all cases of TSTs conversions can be considered false positives due to the classic booster effect or increased cytokine. Psoriasis patients' baseline rates of positive TST results as observed in other studies were close to those found in this study (24%) [39, 40], but the conversion rate was lower even when a booster effect was induced in one month (5% versus 20%) [40, 41]. Considering that three IGRA conversions followed TST conversions, it cannot be ruled out that at least these cases were due to true conversions of anergic tests or recent infection by *Mtb* [34, 41], although no patient developed TB at follow-up. The maintenance of concordance and the absence of difference between tests after treatment suggest that IGRA conversions counteracted the difference in TST conversions.

Some limitations of this study are the lack of an MTX-free control group, chest tomography [42] to improve the diagnosis of LTBI, and cytokine evaluation in skin biopsies to facilitate better correlation with the TST. Studies including larger populations are important to verify the reproducibility of the data.

## Conclusions

In this patient population, TST conversions probably occur by an increase in IFN- γ due to the cytotoxic effect of MTX on psoriasis plaques. However, not all positive tests can be considered false-positive due to cytokine increase or to classical booster effect. In psoriasis patients taking MTX, the responsible for screening should evaluate the possibility of using other techniques, such as the IGRA or chest tomography, concurrently with the TST to increase the accuracy of LTBI diagnosis when TST yields intermediate results (5–10 mm).

## Supporting information

**S1 Checklist. TREND statement checklist.**
(PDF)

**S1 Table. Measures of associations between positive and negative IGRA results and the average values of numerical variables before MTX treatment.**
(DOCX)

**S2 Table. Measures of associations between positive and negative TST results and the average values of numerical variables before MTX treatment.**
(DOCX)

**S3 Table. Measures of associations between positive and negative IGRA results and the average values of numerical variables after MTX treatment.**
(DOCX)

**S4 Table. Measures of associations between positive and negative TST results and the average values of numerical variables after MTX treatment.**
(DOCX)

**S1 Fig. CONSORT 2010 flow diagram.**
(PDF)

**S1 File.**
(DOCX)

## Acknowledgments

The authors thank Dr. Carolina Chacon and Dr. Rui do Rego Barros, who referred patients to our outpatient clinics. We also thank Dr. Rodrigo Castro de Medeiros and Dr. João Paulo Lima, who contributed ideas to the discussion.

## Author Contributions

**Conceptualization:** Vanessa Lucília Silveira de Medeiros, Vera Magalhães.

**Data curation:** Vanessa Lucília Silveira de Medeiros, Lílian Maria Lapa Montenegro, Valdênia Maria Oliveira de Souza.

**Formal analysis:** Fabiana Cristina Fulco Santos, Lílian Maria Lapa Montenegro, Maria da Conceição Silva, Valdênia Maria Oliveira de Souza, Reginaldo Gonçalvez de Lima Neto, Líbia Cristina Rocha Vilela Moura.

**Investigation:** Vanessa Lucília Silveira de Medeiros, Fabiana Cristina Fulco Santos, Lílian Maria Lapa Montenegro, Maria da Conceição Silva, Valdênia Maria Oliveira de Souza, Reginaldo Gonçalvez de Lima Neto, Vera Magalhães.

**Methodology:** Vanessa Lucília Silveira de Medeiros, Maria da Conceição Silva, Valdênia Maria Oliveira de Souza.

**Project administration:** Vanessa Lucília Silveira de Medeiros.

**Resources:** Vanessa Lucília Silveira de Medeiros, Lílian Maria Lapa Montenegro, Valdênia Maria Oliveira de Souza.

**Supervision:** Líbia Cristina Rocha Vilela Moura, Vera Magalhães.

**Validation:** Fabiana Cristina Fulco Santos, Lílian Maria Lapa Montenegro, Valdênia Maria Oliveira de Souza, Reginaldo Gonçalvez de Lima Neto, Líbia Cristina Rocha Vilela Moura, Vera Magalhães.

**Visualization:** Fabiana Cristina Fulco Santos, Lílian Maria Lapa Montenegro, Maria da Conceição Silva, Valdênia Maria Oliveira de Souza, Reginaldo Gonçalvez de Lima Neto, Líbia Cristina Rocha Vilela Moura, Vera Magalhães.

**Writing – original draft:** Vanessa Lucília Silveira de Medeiros.

**Writing – review & editing:** Vanessa Lucília Silveira de Medeiros, Líbia Cristina Rocha Vilela Moura, Vera Magalhães.

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
