## [Decision Letter · Decision Letter 0]

15 Sep 2020

PONE-D-20-23955

TST conversions and an increase in systemic interferon-gamma after MTX introduction in psoriasis patients: An interventional study associated with a cohort.

PLOS ONE

Dear Dr. Medeiros,

Thank you for submitting your manuscript to PLOS ONE. After careful consideration, we feel that it has merit but does not fully meet PLOS ONE’s publication criteria as it currently stands. Therefore, we invite you to submit a revised version of the manuscript that addresses the points raised during the review process.

We look forward to receiving your revised manuscript.

Kind regards,

Remco PH Peters, MD, PhD, DLSHTM

Academic Editor

PLOS ONE

Journal Requirements:

2. Thank you for submitting your clinical trial to PLOS ONE and for providing the name of the registry and the registration number. The information in the registry entry suggests that your trial was registered after patient recruitment began. PLOS ONE strongly encourages authors to register all trials before recruiting the first participant in a study.

i) your reasons for your delay in registering this study (after enrolment of participants started);

ii) confirmation that all related trials are registered by stating: “The authors confirm that all ongoing and related trials for this drug/intervention are registered”.

Please also ensure you report the date at which the ethics committee approved the study as well as the complete date range for patient recruitment and follow-up in the Methods section of your manuscript.

3.  Thank you for including your ethics statement: 'This study was a prospective, quasi-experimental, single-arm, self-controlled study associated with a cohort approved by the ethics committee, with trial number 42859015.0.0000.5208-CAAE on 01/04/2015. The authors confirm that all ongoing and related trials for this drug/intervention are registered at http://www.ensaiosclinicos.gov.br/rg/RBR-2mt78b/ by the number RBR-2mt78b. Forty-five patients were referred to initiate follow-up during the study period and signed the consent form.'

Reviewers' comments:

Reviewer's Responses to Questions

**Comments to the Author**

1. Is the manuscript technically sound, and do the data support the conclusions?

Reviewer #1: Partly

Reviewer #2: Partly

2. Has the statistical analysis been performed appropriately and rigorously? 

Reviewer #1: I Don't Know

Reviewer #2: I Don't Know

3. Have the authors made all data underlying the findings in their manuscript fully available?

Reviewer #1: Yes

Reviewer #2: Yes

4. Is the manuscript presented in an intelligible fashion and written in standard English?

Reviewer #1: Yes

Reviewer #2: Yes

5. Review Comments to the Author

Reviewer #1: Immunosuppression is an important cause of the reactivation of LTBI. Although there are no highly specific diagnostic tests for the diagnosis of LTBI, the use of TST and IGRA are surrogate markers. This study provides prospective data to interpret TST/IGRA conversion in patients with psoriasis who are taking MTX. However, the small sample size makes the data difficult to interpret. Given that participants were BCG vaccinated, the significance of TST changes, while IGRA remains unchanged is hard to interpret. Perhaps the results are a result of boosting (although the repeat test is at 90 days)? I appreciate that the authors have acknowledged several of the limitations of the TST however the sample size is very small, making it difficult to interpret further. I can see that there is a sample size calculation, perhaps a bio-statistician is better placed to comment if conclusions can be made based on the calculation and sample size (not sure if this has already been completed?). It does however, seem the authors have addressed some of the concerns of the previous reviewers.

Although reassuring that no one developed TB in the three-month period, it is a very short follow up period.

- Mix of past and present tense e.g. line 172 (there were no exclusions…), line 241, 317

- Line 206 change to “they were advised to attend the consultation….”

- Line 212 remove “they presented”

- Line 234: should missing data be considered zero or not applicable, as this may artificially increase the number of negative test results (or where these tests not considered negative)?

- Line 276: can this be worded more clearly?

- Line 312: ?”report a discordance between…”

- Line 360/1 – sentence structure needs to be changed

- Sometimes the discussion is a little difficult to follow, one final proofread with a focus on structure may help.

Reviewer #2: In the "Discussion" session, the authors did not mention the possibility of "Boosting effect" occurrence. Even though the TST repetition was performed 12 weeks after the first exam, "Boosting effect" could had happened.

Reference: Cauthen GM, Snider DE Jr, Onorato IM. Boosting of tuberculin sensitivity among Southeast Asian refugees. Am J Respir Crit Care Med. 1994;149(6):1597-1600. doi:10.1164/ajrccm.149.6.8004319

6. PLOS authors have the option to publish the peer review history of their article (what does this mean?). If published, this will include your full peer review and any attached files.

Reviewer #1: No

Reviewer #2: No

---

## [Author Response · Author response to Decision Letter 0]

18 Oct 2020

Response to Reviewers

Answer: I corrected the topics according to the Plos one formatting

2. Thank you for submitting your clinical trial to PLOS ONE and for providing the name of the registry and the registration number. The information in the registry entry suggests that your trial was registered after patient recruitment began. PLOS ONE strongly encourages authors to register all trials before recruiting the first participant in a study.

i) your reasons for your delay in registering this study (after enrolment of participants started);

ii) confirmation that all related trials are registered by stating: “The authors confirm that all ongoing and related trials for this drug/intervention are registered”.

Please also ensure you report the date at which the ethics committee approved the study as well as the complete date range for patient recruitment and follow-up in the Methods section of your manuscript.

Answer:

i) I added the justification in the text

ii) I confirmed that all related trials are registered

3. Thank you for including your ethics statement: 'This study was a prospective, quasi-experimental, single-arm, self-controlled study associated with a cohort approved by the ethics committee, with trial number 42859015.0.0000.5208-CAAE on 01/04/2015. The authors confirm that all ongoing and related trials for this drug/intervention are registered at http://www.ensaiosclinicos.gov.br/rg/RBR-2mt78b/ by the number RBR-2mt78b. Forty-five patients were referred to initiate follow-up during the study period and signed the consent form.'

Answer:

a) I added the name of the ethical committee

b) The ethic statement was added to the field of the submission

Answer: All the ethics were included in a separate part of “Ethical statements” and were removed from the text.

Responses to Questions

Reviewer #1: Immunosuppression is an important cause of the reactivation of LTBI. Although there are no highly specific diagnostic tests for the diagnosis of LTBI, the use of TST and IGRA are surrogate markers. This study provides prospective data to interpret TST/IGRA conversion in patients with psoriasis who are taking MTX. However, the small sample size makes the data difficult to interpret. Given that participants were BCG vaccinated, the significance of TST changes, while IGRA remains unchanged is hard to interpret. Perhaps the results are a result of boosting (although the repeat test is at 90 days)? I appreciate that the authors have acknowledged several of the limitations of the TST however the sample size is very small, making it difficult to interpret further. I can see that there is a sample size calculation, perhaps a bio-statistician is better placed to comment if conclusions can be made based on the calculation and sample size (not sure if this has already been completed?). It does, however, seem the authors have addressed some of the concerns of the previous reviewers.

Although reassuring that no one developed TB in the three months, it is a very short follow up period.

Answer : 

1) The sample was above the minimum required. Studies with paired variables require a smaller sample. At the end of the text, it is written that it is necessary to repeat the research to verify the reproducibility of the results, preferably in larger samples.

2) The follow-up was 18 months for each participant. This is in the methods section line 118-119.

3) Thank you for your consideration. When I wrote false-positive tests, I included the booster effect. The increase in TST size in MTX psoriasis patients was found in other studies in countries with less environmental exposition to Mycobacterium tuberculosis. Psoriasis patients who undergoing re-testing in one month, have 5% conversions (1); Course the booster effect can occur in three months, but it does not explain everything. There would be three possibilities, the classic booster for the repetition of the TST, the increase of the identification for the increase of the INF gamma and TNF-alfa, and some cases of possible true conversions. IGRA would try to identify these true conversions. The best way to evaluate the booster effect was a control group without treatment. It is placed as a limitation. The text was not very clear about this. I make the text more clear and cited the booster effect.

(1) Ribera M, Zulaica A, Pujol C, et al. Estimation of the prevalence of latent tuberculosis infection in patients with moderate to severe plaque psoriasis in Spain: The Latent study. Actas Dermosifiliogr. 2015;106(10):823-829. DOI:10.1016/j.ad.2015.08.001

- Mix of past and present tense e.g. line 172 (there were no exclusions…), line 241, 317 

Thank you for the warning, CORRECTED. English was again revised by experts.

- Line 206 change to “they were advised to attend the consultation….” - CHANGED

- Line 212 remove “they presented” - REMOVED

- Line 234: should missing data be considered zero or not applicable, as this may artificially increase the number of negative test results (or where these tests not considered negative)? 

 ANSWER: Plos requests to be declared how the missing values were handled. Statistically, the missing data can be filled with values close to mean, estimated values (regression imputation), last observation made found, maximum probability, filled with zero / negative and others possibilitys (1) . Considering them equal to zero reduced our associations. However, there were no missing data for IGRA or TST. We only had for TNF-α and ESR for lack of material in the laboratory in the second test (Line 299). This was probably the reason why TNF-α increased without significance. This may be the reason for the loss of association between IGRA and ESR in the second test, but this discussion would be very hypothetical.

(1) Kang H. The prevention and handling of the missing data. Korean J Anesthesiol. 2013;64(5):402-406. doi:10.4097/kjae.2013.64.5.402

- Line 276: can this be worded more clearly? ANSWER: I tried to make it clear

- Line 312: ?” report a discordance between…” CHANGED

- Line 360/1 – sentence structure needs to be changed - CHANGED

- Sometimes the discussion is a little difficult to follow, one final proofread with a focus on the structure may help.

ANSWER

1) The sequence is: IGRA/TST discordance � epidemiological/ clinical factors � Inflammatory process of psoriasis � Influence of immunosuppression � increase in INF-gamma and TNF-alfa/ cytotoxicity as a partial explanation for changes in TST� discussion about being false ou true positives according to IGRA agreement

2) I changed the words always using “disagreement” to mark the concept in this the discussion as you suggested above

3) I made a change in the structure of the paragraph about the cytotoxic effect of MTX to improve text fluency

4) I added the discussion about the booster effect.

Reviewer #2: In the "Discussion" session, the authors did not mention the possibility of "Boosting effect" occurrence. Even though the TST repetition was performed 12 weeks after the first exam, the "Boosting effect" could have happened.

Reference: Cauthen GM, Snider DE Jr, Onorato IM. Boosting of tuberculin sensitivity among Southeast Asian refugees. Am J Respir Crit Care Med. 1994;149(6):1597-1600. DOI:10.1164/ajrccm.149.6.8004319

ANSWER: Thank you for your consideration. When I speak about false positives I include the booster effect. The increase in TST size in MTX psoriasis patients was found in other studies in countries with less environmental exposition to Mycobacterium tuberculosis. Psoriasis patients who undergoing re-testing in one month, have 5% conversions (1); Course the booster effect can occur in three months, but it does not explain everything. There would be three possibilities, the classic booster for the repetition of the TST, the increase of the identification for the increase of the INF gamma and TNF-alfa, and some cases of possible true conversions. IGRA would try to identify these true conversions. The best way to evaluate the booster effect was a control group without treatment. I tried to improve the text.

(1) Ribera M, Zulaica A, Pujol C, et al. Estimation of the prevalence of latent tuberculosis infection in patients with moderate to severe plaque psoriasis in Spain: The Latent study. Actas Dermosifiliogr. 2015;106(10):823-829. DOI:10.1016/j.ad.2015.08.001

---

## [Decision Letter · Decision Letter 1]

27 Oct 2020

TST  conversions and systemic interferon-gamma increase after methotrexate introduction in psoriasis patients

PONE-D-20-23955R1

Dear Dr. Medeiros,

We’re pleased to inform you that your manuscript has been judged scientifically suitable for publication and will be formally accepted for publication once it meets all outstanding technical requirements.

Kind regards,

Remco PH Peters, MD, PhD, DLSHTM

Academic Editor

PLOS ONE

Additional Editor Comments (optional):

Please respond to Reviewer 3's suggestions:

Reviewer 3:

A single arm, self-controlled study was conducted to determine if the results of the tuberculin skin test (TST) or interferon-gamma release assay in psoriasis patients change after 12 weeks of methotrexate treatment. Clinical and inflammatory markers were also investigated. The proportion of positive TST cases significantly increased at post-treatment. Additionally, serum IFN-γ increased at post-treatment.

Minor revisions:

1-  Line 132: Indicate the statistical testing method which achieves 80% power.

2-  Line 212: Indicate the specific type of Wilcoxon test, as there are at least two.

3- Indicate if any adverse events were observed during the course of the study.
---

## [Editor Report · Acceptance letter]

25 Nov 2020

PONE-D-20-23955R1 

TST conversions and systemic interferon-gamma increase after methotrexate introduction in psoriasis patients 

Dear Dr. de Medeiros:

I'm pleased to inform you that your manuscript has been deemed suitable for publication in PLOS ONE. Congratulations! Your manuscript is now with our production department. 

Kind regards, 

on behalf of

Prof Remco PH Peters 

Academic Editor

PLOS ONE